# Physiological and Biochemical Mechanisms of Exogenous Melatonin Regulation of Saline–Alkali Tolerance in Oats

Qiang Wang [1,2,†], Weiwei Xu [1,3,†], Changzhong Ren [1], Chao Zhan [1], Chunlong Wang [1], Junwei Li [1], Qinyong Ren [1,3], Xiaotian Liang [1,3], Liming Wei [1], Dabing Xiang [2], Junying Wang [4,*] and Laichun Guo [1,2,*]

1 Baicheng Academy of Agricultural Sciences, No. 17, Sanhe Road, Taobei District, Baicheng 137099, China; wangqiangeternity@icloud.com (Q.W.); weiwei_xu126@126.com (W.X.); renchangzhong@163.com (C.R.); 18204366627@163.com (C.Z.); 13500830529@139.com (C.W.); 18047150407@163.com (J.L.); renqinyong1@163.com (Q.R.); liangxiaotian@mails.jlau.edu.cn (X.L.); vv2315@126.com (L.W.)
2 Key Laboratory of Coarse Cereal Processing, Ministry of Agriculture and Rural Affairs, College of Food and Biological Engineering, Chengdu University, Chengdu 610106, China; dabing.xiang@163.com
3 Agronomy College, Jilin Agricultural University, 2888 New Town Street, Changchun 130118, China
4 Biotechnology Research Institute of Chinese Academy of Agricultural Sciences, No. 12 Zhong Guan Cun South Street, Beijing 100081, China
* Correspondence: wangjunying@caas.cn (J.W.); guolaichun@126.com (L.G.); Tel.: +86-10-82106112 (J.W.); +86-0436-6990003 (L.G.)
† These authors contributed equally to this work.

**Abstract:** Saline–alkali stress is one of the major factors limiting oat seed germination. The regulatory role of melatonin (MT) as a naturally occurring active substance is well known, but the mechanism of MT-mediated intrinsic physiological regulation of oat seed germination under saline–alkali stress is unclear. Therefore, this study investigated (1) the variability of different MT seed soaking concentrations and times on the germination of oat seeds under saline–alkali stress, and (2) the possible physiological regulatory mechanisms of MT on the germination of oat seeds under saline–alkali stress. The results showed that seed vigor was significantly reduced under saline–alkali stress, and seed germination of oats was significantly inhibited; different concentrations of MT seed soaking treatments improved the germination rate, germination potential, germination index, vigor index, root length, germ length, fresh weight, and dry weight; and, overall, treatment improved seed germination and exhibited the phenomenon of "low promotion and high inhibition". The 100 μmol·L$^{-1}$ MT seed soaking treatment effectively enhanced the activities of seed antioxidant enzymes (SOD, POD, CAT, and APX) and nonenzymatic antioxidants (GSH and AsA), affected the AsA-GSH cycle, and effectively increased the contents of osmoregulatory substances (proline, protein, and soluble sugar) by reducing levels of $H_2O_2$, $OH^-$, and MDA, thus enhancing the tolerance of oats to saline–alkali and promoting seed germination. In conclusion, MT has a positive effect on the saline–alkali tolerance of oat seeds.

**Keywords:** saline–alkali; melatonin; oat; germination; antioxidant enzyme activity

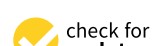



## 1. Introduction

In recent years, with changes in ecological and climatic environmental factors, soil saline–alkalization has become a significant abiotic limiting factor hindering global crop growth and product development and seriously affecting the biosphere and environmental structure [1]. According to reports, more than 900 million hectares of land worldwide are threatened by saline–alkali stress; this high salt and high pH environment hinders the sprouting and growth of crops, causing lower crop yields and endangering world food security [2]. Many studies have shown that mixed soda salt-base stress consisting of both neutral salts such as NaCl and $Na_2SO_4$ and of alkaline salts such as $Na_2CO_3$ and $NaHCO_3$ leads to more severe ionic imbalance, inhibition of antioxidant systems, accumulation

of excess reactive oxygen (ROS), reduction of osmotic substance content, and inhibition of plant seed germination and growth [3,4]. For example, soybean accumulates large amounts of Hydrogen peroxide ($H_2O_2$) and Superoxide anion ($O^{2-}$) under saline stress and reduces Peroxidase (POD) and Reduction-type VC (AsA) concentrations [5]. The ROS metabolism of maize seedlings subjected to saline–alkali stress is inhibited and induces the onset of membrane peroxidation, triggering the inactivation of antioxidant enzymes and thus inhibiting seedling growth [6]. Therefore, finding new regulatory substances to eliminate ROS is significant to protect the normal germination and growth of plants under unfavorable conditions.

As natural active substances, plant growth regulators have been widely used in modern agricultural production because of their essential role in regulating plant seed germination, growth and development, flowering, and fruit set, as well as improving plant resistance.

Melatonin (MT) is a biomolecule derived from tryptophan and is present in almost all living organisms. Melatonin is a potent antioxidant in plants that maintains the balance of ROS metabolism in plant cells. In addition to its regulatory effects on plant seed germination, seedling growth, leaf senescence, and fruit ripening [7], it also enhances plant adaptation to abiotic stresses such as high temperature, drought, cold, salinity, and biotic stresses such as pathogenic bacteria [8,9]. These conclusions were confirmed by the study of Arnao and Hernandez-Ruiz [10], which suggested that stressful environments can induce endogenous melatonin biosynthesis and prompt plant stress responses to abiotic and biotic stresses. Related studies have shown that plants can absorb melatonin from the external environment through seeds, roots, and leaves [9,11,12]. For practical applications, the exogenous application of melatonin to repair stress damage has been reported in various plants, including rice [13], beet [14], cotton [15], and small rye [16]. Several studies have demonstrated that applying exogenous MT can promote seed germination and improve plant resistance to stress. A previous study on cotton found that exogenous MT could enhance its salt tolerance and improve seed germination by regulating Abscisic acid (ABA) and Axillary bud outgrowth (IAA) biosynthesis and by mediating the expression of hormone-related genes [17]. There is evidence that MT application increases seed antioxidant enzyme activity while protecting the cell morphology from oxidative damage to the seeds [16,18]. Under salt stress, the Superoxide dismutase (SOD), POD, and Catalase (CAT) activities of MT-pretreated cucumber seeds increased 1.3–5.0-fold compared with untreated seeds [19]. Similarly, MT treatment reduced Malondialdehyde (MDA) and $H_2O_2$ concentrations in maize seeds and enhanced SOD, POD, CAT, and AsA activities to prevent cellular oxidative damage [20]. Exogenous MT application reduced $H_2O_2$, $O^{2-}$, and MDA contents in seeds under abiotic stress while increasing their range of organic osmolytes and enhancing the cellular scavenging of ROS [7]. These physiological effects of exogenous MT on seed germination show a phenomenon of "low promotion and high inhibition", which is closely related to application concentration and duration [21,22]. Therefore, further development of MT physiological functions is of vast significance for agricultural production.

Oat, also known as wild wheat, oat, and oat grass, is an annual herb grown in full sun. As a forage crop, oat forage has the advantages of high nutrient content, high digestibility, and good palatability; as a mixed crop, it contains protein, fat, starch, glucan, and other nutrients, and its nutritional value is better than that of sorghum, barley, corn, and other crops [23]. Oat also has a variety of physiological functions such as lowering blood sugar and blood lipids, reducing intestinal inflammation, and regulating blood pressure, and it is used in ecological restoration for sand control and soil consolidation, improving saline soil and other purposes [24]. However, abiotic stresses such as drought and saline–alkali conditions directly inhibit seed germination and the growth and development of oats. For example, with increasing saline–alkali levels, oat root biomass and the root–shoot ratio are significantly decreased, and the cation uptake coefficient is decreased at the filling stage [25]. Oat yield decreased by approximately 70% under saline–alkali stress [26].

Zhang et al. [27] similarly found that saline–alkali stress inhibited photosynthesis in oat leaves, significantly reducing yield. Previous studies on saline–alkali tolerance in oats have focused on germplasm selection. Therefore, it is necessary to identify new saline–alkali tolerance strategies to improve the tolerance of oats to saline–alkali stress.

Several previous studies have confirmed that exogenous MT can promote seed germination under abiotic stress, stimulate seedlings to resist stress, and alleviate the inhibition of seed germination by stressful environments. However, the specific mechanism of action of exogenous MT treatment on oat seed germination under soda saline–alkali stress needs to be more precisely understood. Therefore, the objectives of this study were (1) to determine the effects of differences in exogenous MT seed soaking concentration and time on oat seed germination under saline–alkali stress, and (2) to investigate the possible physiological and biochemical mechanisms of MT seed soaking treatment in promoting oat seed germination under saline–alkali stress. The aim is to broaden the scope of ideas for oat saline–alkali tolerance research and provide new plant saline–alkali tolerance strategies.

## 2. Materials and Methods

### 2.1. Experimental Materials

This experiment was conducted with the saline–alkali sensitive variety of naked oats "Baiyan No.5", cultivated and provided by the Academy of Agricultural Sciences of Baicheng City, Jilin Province. The seeds were stored in a dry environment at 4 °C before the start of the experiment.

### 2.2. Solution Preparation

Due to the complex and variable characteristics of salinity and alkalinity of saline land in the central and western regions of Jilin Province, the saline–alkali mixture was configured with a modified 50% Hoagland nutrient solution (nutrient composition as shown in Appendix A Table A1) in the molar ratio of 1:9:1:9 of $NaCl:Na_2SO_4:Na_2CO_3:NaHCO_3$ to simulate the local saline–alkali soil environment.

MT is easily decomposed in the presence of light, so it was prepared immediately before use. MT (0.116 g, analytically pure) was dissolved in 3 mL of anhydrous ethanol, deionized water was added, and the volume was adjusted to 250 mL to obtain a 2 mmol·$L^{-1}$ MT stock solution. The MT solution was diluted with deionized water to 25, 50, 75, 100, 150, and 200 μmol·$L^{-1}$. After preparation, the sample was stored at room temperature away from light, and 0.05% (*v/v*) Tween-20 was added.

### 2.3. Experiment 1: Effect of MT on Seed Germination of Oats under Soda Saline–Alkali Stress

The experiment was conducted in 2022 in the greenhouse of the Academy of Agricultural Sciences, Baicheng City, Jilin Province (122°48′1″ E, 45°36′56″ N, 157.51 m above sea level). Seeds with complete and uniform grains and no pests were selected, disinfected with 75% alcohol for 30 s, washed 3–5 times with distilled water for 2 min each time, disinfected in 1% sodium hypochlorite solution for 8 min, rinsed 3–5 times with distilled water, and placed in a dark place to air dry. The sterilized seeds were soaked in a 0, 25, 50, 75, 100, 150, or 200 μmol·$L^{-1}$ MT solution in a constant temperature incubator at 25 °C for 12 or 24 h. The soaked treated seeds were rinsed 2–3 times with distilled water, and blotting paper was used to absorb the surface water. Seeds with uniform dewlap were selected and placed evenly in Petri dishes (φ9 cm) lined with two layers of sterile filter paper with 30 seeds per dish, and each treatment was replicated three times. Seven milliliters of a 45 μmol·$L^{-1}$ saline–alkali solution (preliminary experiments showed that this concentration was the median lethal concentration for oat seed germination, with pH 8.97, conductivity 11.04 mS/cm, and salinity 6.18 PSU) was added to each Petri dish separately, and 50% Hoagland nutrient solution was added as a control (CK). Petri dishes were placed on a greenhouse germination bed, illuminated using red, blue, and white light (red: blue: white = 3:2:1), and incubated with alternating light/dark (16 h/8 h) at 20 ± 1 °C and 40 ± 5% humidity. The water that dissipated from the Petri dishes was replenished using the weighing method,

and the saline–alkali solution was replaced every 2 d during 7 d of incubation. There were 16 treatments in the experiment: S45 + T12~24, 45 µmol·L$^{-1}$ saline–alkali solution culture + 0 µmol·L$^{-1}$ MT dip for 12 h or 24 h; S45 + MT25~200 + T12~24, 45 µmol·L$^{-1}$ saline–alkali solution culture + 25~200 µmol·L$^{-1}$ MT dip for 12 h or 24 h; and CK, seed soaking in distilled water for 12 h or 24 h + 50% of Hoagland nutrient solution culture, which was used as a control.

### 2.3.1. Seed Germination Index

According to the "International Seed Inspection Requirements", the number of germinated seeds in each treatment group was counted every 12 h from the time the seeds were placed in the greenhouse germination bed for 7 d (germination was considered to have occurred when the radicle of the seeds broke through the seed coat by 2 mm) [7].

$$\text{Germination rate (GR)} = \text{number of germinated seeds in the first 7 d/number of seeds for testing} \times 100\%$$

$$\text{Germination potential (GP)} = \text{number of seeds germinated in the first 4 d/number of seeds for testing} \times 100\%$$

$$\text{Germination index (GI)} = \sum (Gt/Dt)$$

$$\text{Vigor Index (VI)} = GI \times S$$

where Gt is the number of seeds germinated at time t (d), Dt is the corresponding germination days, and S is the radicle length.

### 2.3.2. Morphological and Biomass Indicators

Ten seedlings of uniform growth (stripped seed coat) were selected from each treatment group, and their root length (RL) and germ length (GL, the linear distance from the base of the radicle to the top of the leaf) were measured separately with a Vernier caliper and a straightedge. After the root system and seedling height measurements were completed, the plant surface was washed with deionized water, absorbent paper was used to remove surface water, and then the fresh weight (FW) was immediately recorded with an electronic balance. The tissues were then killed at 105 °C for 30 min, dried at 80 °C for 72 h until the mass was constant, and weighed to record the dry weight (DW).

### 2.3.3. A Comprehensive Evaluation of Different MT Treatments to Improve the Saline–Alkali Tolerance of Oats

The analysis of the saline–alkali tolerance of oats using a single index is somewhat one-sided; fuzzy function affiliation function analysis [28] can overcome the shortcomings caused by a small number of evaluation indices. The evaluation system is calculated as follows:

$$Y_{ij} = \frac{X_{ij} - X_{\min}}{X_{\max} - X_{\min}}$$

$$Y_{ij}{}' = 1 - \frac{X_{ij} - X_{\min}}{X_{\max} - X_{\min}}$$

$$\overline{X_i} = \frac{1}{n} \sum_{j=1}^{n} X_{ij}$$

$$V_j = \frac{\sqrt{\sum_{j=1}^{n} \left( X_{ij} - X_i^- \right)}}{X_j^-}$$

$$W_i = \frac{V_i}{\sum_{i=1}^{m} V_i}$$

$$D_j = \sum_{i=1}^{n} \left( W_i \times Y_{ij} \right)$$

where $Y_{ij}$ is the value of the affiliation function of the *j*th treatment of the *i*th indicator, $X_{ij}$ is the measured value of the indicator for the *j*th treatment of the *i*th hand, $X_{\min}$ is the minimum value of the measured indicator, $X_{\max}$ is the maximum value among the measured indicators, $Y_{ij}'$ is the value of the inverse affiliation function of the *j*th treatment of the *i*th hand, $\overline{X_i}$ is the mean value of the *i*th indicator, *n* is the number of treatments, *m* is the number of indicators, $V_j$ is the standard deviation coefficient of the *i*th indicator, $W_i$ is the weight of the *i*th indicator, and $D_j$ is the combined score value of the *j*th treatment.

### 2.4. Experiment 2: Study of the Intrinsic Physiological Mechanism of Oat Seed Germination under Saline–Alkali Stress and MT Treatment

The seed disinfection treatment and culture environment were the same as in Experiment 1. The sterilized seeds were soaked in a 100 µmol·L$^{-1}$ MT solution with an equal amount of distilled water as the control and incubated in a constant dark incubator at 25 °C for 24 h. The seeds were rinsed with distilled water 2–3 times after soaking, and absorbent paper was used to dry the surface water. Dewy uniform seeds were selected and evenly placed in Petri dishes (φ17 cm) lined with two layers of sterile filter paper and exposed to saline–alkali stress using a 45 mmol·L$^{-1}$ saline–alkali solution. Each Petri dish was spiked with 16 mL of saline–alkali solution with 120 grains per dish, and with an equal amount of 50% Hoagland nutrient solution as a control, and each treatment was replicated three times with five Petri dishes per replicate for a total of 45 Petri dishes. The experimental treatments were as follows: CK, seeds water soaked for 24 h + 50% Hoagland nutrient solution culture; S45, seeds water soaked for 24 h + 45 mmol·L$^{-1}$ saline–alkali solution culture; and S + MT, seeds 100 µmol·L$^{-1}$ MT soaked for 24 h + 45 mmol·L$^{-1}$ saline–alkali solution culture. The seeds were sampled at 1 d, 3 d, and 5 d. Each time the seeds were tested, seeds with consistent and representative growth were selected from the 5 Petri dishes of each replicate, mixed and sampled, and then stored in a −80 °C refrigerator after quick freezing with liquid nitrogen for the subsequent determination of physiological indices.

#### 2.4.1. Antioxidant Enzyme Activity

The kits were provided by Suzhou Michy Biomedical Technology Co., Ltd. (Suzhou, China). Superoxide dismutase (SOD, M0102A) activity was determined by the WST-8 method [29]. Peroxidase (POD, M0105A) was determined by the guaiacol method [30]. Catalase (CAT, M0103A) was determined by the UV absorption method [31]. Ascorbic peroxidase (APX, M0403A) was determined by the UV-absorption method [32].

#### 2.4.2. Nonenzymatic Antioxidant Activity

Reduced glutathione (GSH) was measured by the DTNB method [33] and reduced-type VC (AsA) was measured by the red phenanthroline method [34] using the kits provided by Suzhou Michy Biomedical Technology Co., Ltd.

#### 2.4.3. MDA Content

Malondialdehyde (MDA) content was determined by the thiobarbituric acid method [35] using a kit (M0106A) provided by Suzhou Michy Biopharmaceutical Technology Co., Ltd.

#### 2.4.4. Osmoregulatory Substances

The soluble sugar content was determined by the anthrone colorimetric method [36], the proline content was determined by the ninhydrin method [37], and the protein con-

tent was determined by the BCA method [38] using the kits provided by Suzhou Michy Biomedical Technology Co., Ltd.

### 2.4.5. $H_2O_2$ Content and $OH^-$ Scavenging Capacity

The $H_2O_2$ content scavenging ability was determined by the titanium sulfate method [39] and $OH^-$ content was determined by the salicylic acid method [40] using kits provided by Suzhou Michy Biopharmaceutical Technology Co., Ltd.

### 2.5. Statistical Analysis

The raw data were collated and analyzed using Excel 2021 (Microsoft Crop, Montgomery, AL, USA), and ANOVA and LSD tests were performed using SPSS 24.0 (IBM, Chicago, AL, USA). Duncan's multiple extreme difference test was used to measure significant differences between samples ($p < 0.05$). GraphPad Prism 9.4 (GraphPad Software Crop.) was used for plotting, and the correlation analysis was performed at https://www.chiplot (accessed on 6 April 2023). Online correlation analysis was performed to draw heatmaps. The hierarchical clustering analysis of physiological indicators of oat seeds was performed in MetaboAnalyst online software (http://www.metab oanal yst.ca/ (accessed on 9 April 2023)).

### 3. Results

#### 3.1. Effect of Different Concentrations of MT Seed Soaking Treatment on Seed Germination of Oats under Saline–Alkali Stress

#### 3.1.1. Effect on Seed Germination Indicators

As shown in Table 1, the GR, GP, GI, and VI of oat seeds were all significantly improved ($p < 0.05$) to different degrees after different concentrations of MT seed soaking treatment compared with those of the S45 group and the treatment effect under the 100 µmol·L$^{-1}$ treatment. The GR, GP, GI, and VI were increased by 42.55%, 47.73%, 44.69%, and 85.65%, respectively, after 24 h of 100 µmol·L$^{-1}$ MT treatment compared with those of the S45 group.

**Table 1.** Effect of different MT seed soaking concentrations and times on the germination parameters of oat seeds under saline–alkali stress.

| Treatment | | GR (%) | GP (%) | GI | VI |
|---|---|---|---|---|---|
| | CK | 78.89 ± 1.92 | 68.89 ± 3.85 | 40 ± 2.39 | 502.3 ± 33.31 |
| | S45 + MT25 | 52.22 ± 1.92 | 45.56 ± 3.85 | 21.73 ± 0.44 | 217.67 ± 19.38 |
| | S45 + MT25 | 62.22 ± 3.85 ns | 54.44 ± 1.92 ns | 28.43 ± 0.67 ns | 307.24 ± 17.82 ns |
| T12 | S45 + MT50 | 47.78 ± 9.62 ns | 38.89 ± 10.72 ns | 19.55 ± 4.45 ns | 192.9 ± 63.51 ns |
| | S45 + MT75 | 63.33 ± 6.67 ns | 56.67 ± 5.77 ns | 29.49 ± 1.89 ns | 297.8 ± 15.02 ns |
| | S45 + MT100 | 70 ± 3.33 ns | 67.78 ± 1.92 ns | 34.62 ± 1.39 [#] | 389.06 ± 14.45 [#] |
| | S45 + MT150 | 58.89 ± 5.09 ns | 53.33 ± 0 ns | 28.92 ± 1.85 ns | 319.05 ± 27.87 ns |
| | S45 + MT200 | 57.78 ± 3.85 ns | 50 ± 12.02 ns | 25.06 ± 5.1 ns | 252.8 ± 67.58 ns |
| | CK | 86.67 ± 5.77 | 85.56 ± 7.7 | 46.55 ± 4.51 | 555.13 ± 56.97 |
| | S45 + MT0 | 52.22 ± 8.39 | 48.89 ± 11.71 | 22.7 ± 4.36 | 190.67 ± 42.77 |
| | S45 + MT25 | 71.11 ± 16.44 ns | 68.89 ± 15.4 ns | 33.7 ± 6.98 ns | 363.25 ± 83.56 * |
| T24 | S45 + MT50 | 71.11 ± 12.62 ns | 64.44 ± 9.62 ns | 29.11 ± 4.46 ns | 308.45 ± 50.94 ns |
| | S45 + MT75 | 66.67 ± 3.33 ns | 63.33 ± 6.67 ns | 29.88 ± 2.39 ns | 320.41 ± 37.37 ns |
| | S45 + MT100 | 85.56 ± 6.94 *** | 83.33 ± 8.82 ** | 38.44 ± 5.38 *** | 453.75 ± 71.55 **** |
| | S45 + MT150 | 74.44 ± 1.92 ns | 72.22 ± 3.85 ns | 32.85 ± 1.8 ns | 353.98 ± 15.39 * |
| | S45 + MT200 | 76.67 ± 10 * | 75.56 ± 8.39 * | 36.05 ± 6.18 * | 383.65 ± 83.1 ** |

Among them, CK is the control; S45 + MT0, 25, 50, 75, 100, 150, and 200 are different concentrations of MT seed soaking followed by 45 mmol·L$^{-1}$ saline–alkali solution culture; T12 and 24 are different MT seed soaking times. GR: germination rate; GP: germination potential; GI: germination index; VI: vigor index. Different lowercase letters after the data in each table column represent the significant differences between treatments. The significant differences between MT treatment and S45 treatment groups at different concentrations are indicated by # (12 h of seed immersion), * (24 h of seed immersion). Among them, [#]—significant ($p \leq 0.05$); ****, ***, **—significant ($p \leq 0.01$); *—significant ($p \leq 0.05$); ns—not significant ($p > 0.05$) ($p < 0.05$).

### 3.1.2. Effect on Morphological Biomass

After different concentrations of MT seed soaking treatment (Table 2), the GL, RL, FW, and DW of oat seeds all increased to different degrees compared with those of the S45 group, and the treatment effect of 24 h seed soaking was greater than that of 12 h, with the strongest effect under the 100 $\mu mol \cdot L^{-1}$ treatment. GL, RL, FW, and DW were increased by 28.74%, 94.95%, 31.73%, and 23.28%, respectively, after 24 h of 100 $\mu mol \cdot L^{-1}$ MT treatment compared with those of the S45 group.

**Table 2.** Effect of different MT seed soaking concentrations and times on the morphology and biomass of oats under saline–alkali stress.

| Treatment | | GL (cm) | RL (cm) | FW (mg) | DW (mg) |
|---|---|---|---|---|---|
| | CK | 12.55 ± 0.09 | 7.29 ± 0.34 | 162.93 ± 3.29 | 14.9 ± 0.42 |
| | S45 + MT0 | 10.01 ± 0.68 | 2.9 ± 0.58 | 111.99 ± 6.69 | 12.25 ± 0.59 |
| | S45 + MT25 | 10.81 ± 0.55 ns | 3.67 ± 0.52 ns | 120.15 ± 6.55 ns | 13.65 ± 0.75 ns |
| T12 | S45 + MT50 | 9.7 ± 1.23 ns | 3.07 ± 0.96 ns | 102.09 ± 18.49 ns | 11.72 ± 2.17 ns |
| | S45 + MT75 | 10.11 ± 0.51 ns | 3.83 ± 0.98 ns | 108.68 ± 5.42 ns | 12.55 ± 0.53 ns |
| | S45 + MT100 | 11.24 ± 0.03 ns | 4.83 ± 0.08 ns | 123.35 ± 4.08 ns | 13.65 ± 0.47 ns |
| | S45 + MT150 | 11.02 ± 0.31 ns | 4.17 ± 0.2 ns | 114.51 ± 4.18 ns | 13.35 ± 0.57 ns |
| | S45 + MT200 | 10.01 ± 0.65 ns | 3.26 ± 0.51 ns | 104.64 ± 0.74 ns | 12.32 ± 0.03 ns |
| | CK | 11.92 ± 0.07 | 8.45 ± 1.8 | 150.26 ± 13.68 | 14.17 ± 1.3 |
| | S45 + MT0 | 8.37 ± 0.47 | 2.91 ± 0.74 | 90.58 ± 2.74 | 10.22 ± 0.29 |
| | S45 + MT25 | 10.75 ± 0.24 *** | 6.11 ± 0.4 ** | 120.8 ± 5.94 ** | 12.75 ± 0.54 * |
| T24 | S45 + MT50 | 10.58 ± 0.16 *** | 5.41 ± 1.29 * | 126.64 ± 8.11 *** | 13.23 ± 0.43 ** |
| | S45 + MT75 | 10.7 ± 0.43 *** | 5.32 ± 0.18 * | 120.4 ± 6.43 ** | 12.42 ± 0.43 ns |
| | S45 + MT100 | 11.79 ± 0.21 **** | 6.06 ± 0.11 ** | 146.19 ± 4.63 **** | 14.48 ± 0.23 **** |
| | S45 + MT150 | 10.78 ± 0.16 **** | 5.67 ± 0.48 ** | 119.31 ± 0.44 ** | 12.6 ± 0.12 * |
| | S45 + MT200 | 10.58 ± 0.59 **** | 5.16 ± 0.86 ns | 124.45 ± 4.13 *** | 12.01 ± 0 *** |

Among them, CK is the control; S45 + MT0, 25, 50, 75, 100, 150, and 200 are different concentrations of MT seed soaking followed by 45 $mmol \cdot L^{-1}$ saline–alkali solution culture; T12 and 24 are different MT seed soaking times. GL: germ length; RL: root length; FW: fresh weight; DW: dry weight. Different lowercase letters after the data in each table column represent the significant differences between treatments. The significant differences between MT treatment and S45 treatment groups at different concentrations are indicated by # (12 h of seed immersion), * (24 h of seed immersion). Among them, ****, ***, **—significant ($p \leq 0.01$), *—significant ($p \leq 0.05$); ns—not significant ($p > 0.05$) ($p < 0.05$).

### 3.1.3. A Comprehensive Evaluation of the Saline–Alkali Tolerance of Oat Seeds during Germination

After the quantitative transformation of the raw test data, pure numbers in the interval of 0–1 were obtained so that the evaluation indices were comparable (the indices with a negative correlation with saline–alkali tolerance are calculated by the inverse affiliation function). The values of the affiliation function of each index under each treatment were summed by combining the weights, and the saline–alkali tolerance of oat seed germination under different MT seed soaking concentration and seed soaking time treatments were evaluated comprehensively based on the size of the D value. A higher D value indicated that the oat seeds were more tolerant of saline–alkali conditions. In Table 3, the D value represents the saline–alkali tolerance of different MT seed soaking treatments on oat seed germination. The comprehensive evaluation showed that S45 + MT100 + T24 treatment was the best; 100 $\mu mol \cdot L^{-1}$ MT seed soaking for 24 h significantly improved the saline–alkali tolerance of oat seeds.

**Table 3.** Affiliation function analysis of the effect of different MT treatment concentrations and times on different indicators of oats under soda saline–alkali stress.

| Treatment | GR | GP | GI | VI | GL | RL | FW | DW | D Value | Rank |
|---|---|---|---|---|---|---|---|---|---|---|
| S45 + MT0 + T12 | 0.12 | 0.15 | 0.12 | 0.10 | 0.48 | 0.00 | 0.39 | 0.48 | 0.20 | 12 |
| S45 + MT0 + T24 | 0.12 | 0.23 | 0.17 | 0.00 | 0.00 | 0.00 | 0.00 | 0.00 | 0.06 | 14 |
| S45 + MT25 + T12 | 0.38 | 0.35 | 0.47 | 0.44 | 0.71 | 0.24 | 0.53 | 0.80 | 0.47 | 8 |
| S45 + MT25 + T24 | 0.62 | 0.68 | 0.75 | 0.66 | 0.70 | 1.01 | 0.54 | 0.59 | 0.65 | 3 |
| S45 + MT50 + T12 | 0.00 | 0.00 | 0.00 | 0.01 | 0.39 | 0.05 | 0.21 | 0.35 | 0.08 | 13 |
| S45 + MT50 + T24 | 0.62 | 0.57 | 0.51 | 0.45 | 0.65 | 0.79 | 0.65 | 0.71 | 0.57 | 5 |
| S45 + MT75 + T12 | 0.41 | 0.40 | 0.53 | 0.41 | 0.51 | 0.30 | 0.33 | 0.55 | 0.41 | 10 |
| S45 + MT75 + T24 | 0.50 | 0.55 | 0.55 | 0.49 | 0.68 | 0.77 | 0.54 | 0.52 | 0.53 | 6 |
| S45 + MT100 + T12 | 0.59 | 0.65 | 0.80 | 0.75 | 0.84 | 0.61 | 0.59 | 0.80 | 0.69 | 2 |
| S45 + MT100 + T24 | 1.00 | 1.00 | 1.00 | 1.00 | 1.00 | 1.00 | 1.00 | 1.00 | 0.99 | 1 |
| S45 + MT150 + T12 | 0.29 | 0.32 | 0.50 | 0.49 | 0.78 | 0.40 | 0.43 | 0.73 | 0.46 | 9 |
| S45 + MT150 + T24 | 0.71 | 0.75 | 0.70 | 0.62 | 0.71 | 0.88 | 0.52 | 0.56 | 0.64 | 4 |
| S45 + MT200 + T12 | 0.26 | 0.25 | 0.29 | 0.24 | 0.48 | 0.11 | 0.25 | 0.49 | 0.27 | 11 |
| S45 + MT200 + T24 | 0.76 | 0.83 | 0.87 | 0.73 | 0.65 | 0.72 | 0.61 | −2.4 | 0.50 | 7 |
| Wi | 0.14 | 0.13 | 0.08 | 0.31 | 0.04 | 0.03 | 0.19 | 0.07 | | |

Among them, CK is the control; S45 + MT0, 25, 50, 75, 100, 150, and 200 are different concentrations of MT seed soaking followed by 45 mmol·L$^{-1}$ saline–alkali solution culture; T12 and 24 are different MT seed soaking times. GR: germination rate; GP: germination potential; GI: germination index; VI: vigor index; GL: germ length; RL: root length; FW: fresh weight; DW: dry weight.

### 3.2. Effect of MT Seed Soaking Treatment on Physiological Indicators of Oats under Saline–Alkali Stress

#### 3.2.1. Effect on the Activity of Antioxidant Enzymes

The changes in SOD, POD, CAT, and APX were measured at different germination periods. MT seed soaking treatment significantly enhanced the activities of SOD (Figure 1a), POD (Figure 1b), CAT (Figure 1c), and APX (Figure 1d) enzymes in oat seeds. After 1 d of saline–alkali stress, oat SOD, POD, CAT, and APX activities were increased in the S + MT group compared with those of the S45 group, and SOD, POD, and CAT significantly increased by 100.48%, 45.48%, and 38.76%, respectively ($p < 0.05$); APX activities were not significantly different ($p > 0.05$). After 3 d of saline–alkali stress, SOD and CAT activities of the S45 group were relatively decreased compared with those of d 1. SOD, POD, and APX activities of the S + MT group were further increased compared with those of d 1. SOD and CAT activities were significantly increased by 195.00% and 160.63%, respectively ($p < 0.05$), compared with those of the S45 group, and there were no significant differences for POD and APX. After 5 d of saline–alkali stress, overall, POD, CAT, and APX activities in the S45 group continued to increase compared with those of the previous group, whereas SOD activity decreased; SOD, POD, and CAT activities in the S + MT group were significantly enhanced compared with those of the S45 group, increasing by 162.02%, 34.57%, and 36.47%, respectively. There was no significant difference in APX activity.

#### 3.2.2. Effect on the Activity of Nonenzymatic Antioxidants

MT seed soaking treatment significantly increased the GSH (Figure 2a) and AsA (Figure 2b) of oat seedlings under saline–alkali stress. After 1 d of saline–alkali stress, the GSH and AsA contents were increased in the S + MT group compared with those of the S45 group, but the effect was not significant ($p > 0.05$). After 3 d of saline–alkali stress, in the S + MT-treated group, the GSH and AsA contents increased significantly by 111.20% and 42.86%, respectively, compared with those of the S45 group ($p < 0.05$). After 5 d of saline–alkali stress, the GSH and AsA contents of the S + MT treatment steadily increased compared with the previous levels; the GSH content increased by 77.69% compared with that of the S45 group, and the AsA content also increased, but the difference was not significant. This indicates that MT promotes the AsA-GSH cycle by enhancing AsA and GSH activities to effectively scavenge ROS and mitigate the damage to the cell membrane by adverse stress.

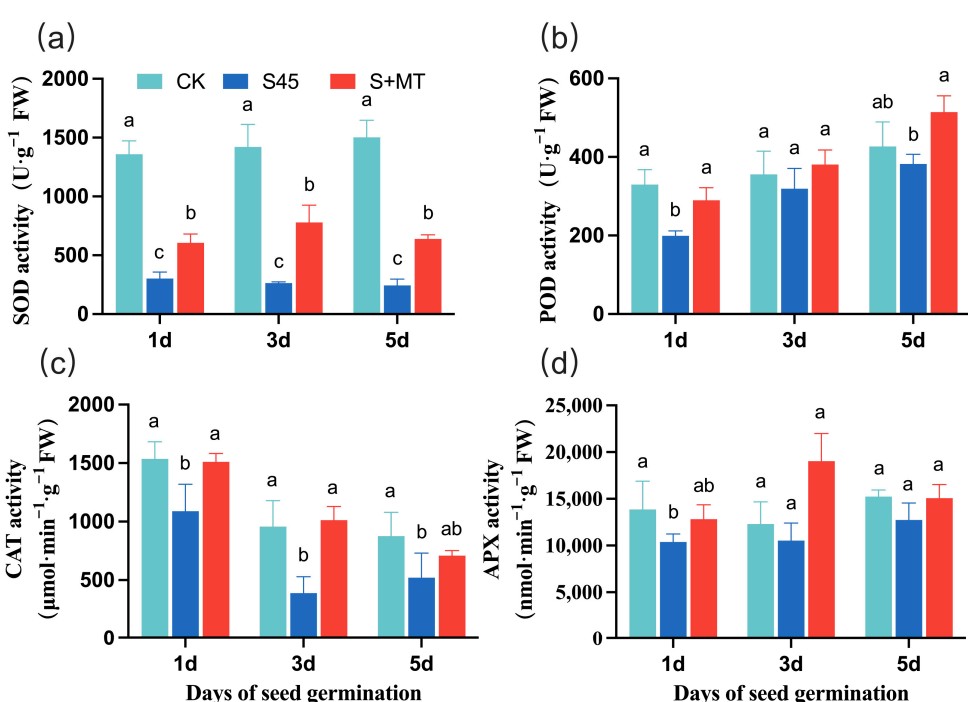

**Figure 1.** Effect of MT seed soaking treatments on the activities of SOD (**a**), POD (**b**), CAT (**c**), and APX (**d**) in oats under saline–alkali stress. CK is the control, S45 is saline–alkali stress, and S + MT is MT treatment + saline–alkali stress. Each lowercase letter represents the significance level of the difference between treatments ($p < 0.05$).

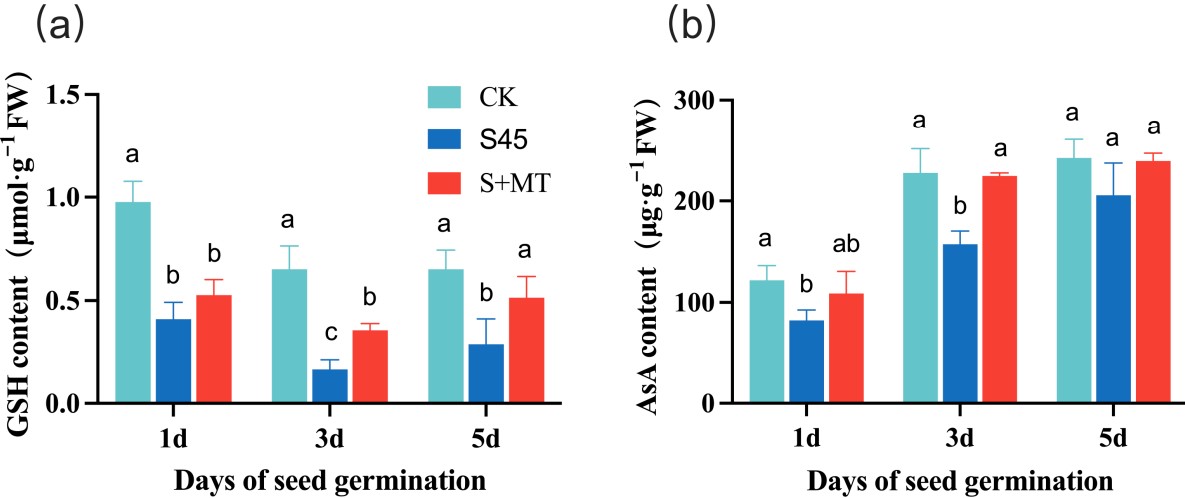

**Figure 2.** Effect of MT seed soaking treatment on GSH (**a**) and AsA; (**b**) activities of oats under saline–alkali stress. CK is the control, S45 is saline–alkali stress, and S + MT is MT treatment + saline–alkali stress. Each lowercase letter represents the significance level of the difference between treatments ($p < 0.05$).

### 3.2.3. Effects on Osmoregulatory Substances and the Degree of Membrane Peroxidation

Saline–alkali stress decreased the protein (Figure 3a), soluble sugar (Figure 3b), and proline contents (Figure 3c) in oats, and the protein, soluble sugar, and proline contents increased significantly ($p < 0.05$) after MT seed soaking treatment and showed a gradual increase with the extension of saline–alkali stress time. After 1 d of saline–alkali stress, the protein concentration in the S45 group was significantly reduced by 29.53% compared with that of CK, whereas the proline content increased by 11.83% compared with that of CK; the proline content was significantly reduced by 33.12%, and the soluble sugar content increased by 20.67% compared with those of the S45 group after MT soaking seed treatment,

whereas there was no significant effect on the protein concentration ($p > 0.05$). After 3 d of saline–alkali stress, the protein concentration, soluble sugar, and proline contents of the S + MT group increased by 36.70%, 98.15%, and 49.56%, respectively, compared with those of the S45 group. After 5 d of saline–alkali stress, the proline content decreased compared with the previous time and increased by 75.26% in the S + MT group compared with that of the S45 group; the protein concentration increased significantly by 54.51%, and the soluble sugar content increased by 76.10% under the S + MT treatment compared with those of the S45 treatment.

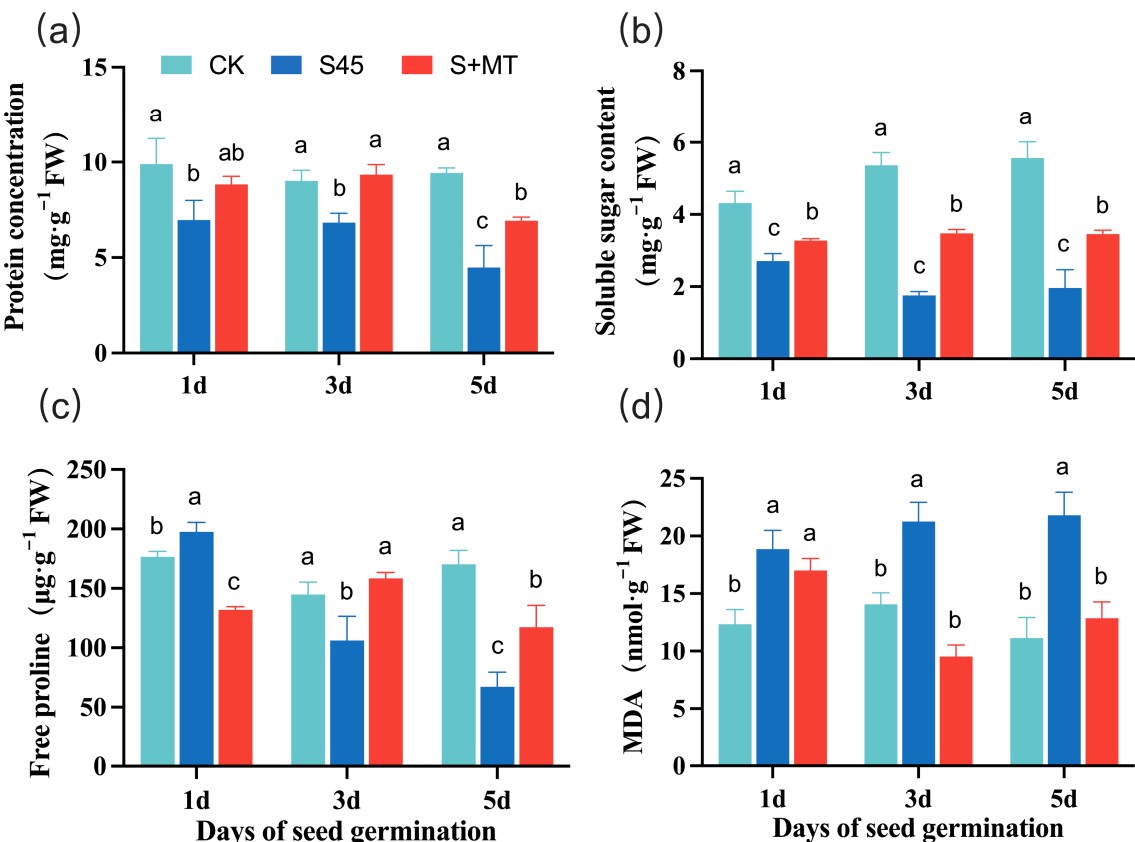

**Figure 3.** Effect of MT seed soaking treatment on the protein (**a**), soluble sugar (**b**), proline (**c**), and MDA (**d**) contents of oats under saline–alkali stress. CK is the control, S45 is saline–alkali stress, and S + MT is MT treatment + saline–alkali stress. Each lowercase letter represents the significance level of the difference between treatments ($p < 0.05$).

MT alleviated membrane lipid peroxidation caused by saline–alkali stress by reducing the MDA content (Figure 3d). The MDA content of the S45 group was higher than that of the CK and S + MT groups under different stress days, and the MDA content showed a gradual increase with an increasing number of stress days. After 1 d of saline–alkali stress, the MDA content of the S45 and S + MT groups differed significantly from that of CK, increasing by 52.97% and 37.84%, respectively; there was no significant difference between the S45 and S + MT groups. At 3 d and 5 d after saline–alkali stress, the MDA content was reduced by 55.17% and 40.98%, respectively, under the S + MT group treatment compared with that of the S45 group ($p < 0.05$).

3.2.4. Effect on ROS Levels

$OH^-$ scavenging capacity can be an important indicator of a substance's antioxidant capacity. Saline–alkali stress significantly increased oat $OH^-$ levels (Figure 4a), whereas the MT seed soaking treatment significantly decreased the $OH^-$ production rate. After 1 d of saline–alkali stress, the $OH^-$ content of the S + MT group was reduced by 78.88%

compared with that of the S45 group, and the difference was significant ($p < 0.05$). After 3 d of saline–alkali stress, the OH$^-$ content of the S45 group was reduced compared with that of the previous time, and the difference between the S + MT and S45 groups was insignificant ($p > 0.05$). After 5 d of saline–alkali stress, the OH$^-$ content of the S + MT group was further reduced compared with the previous time, and the OH$^-$ content of oats in the S + MT treatment was significantly reduced by 63.03% compared with that of the S45 treatment.

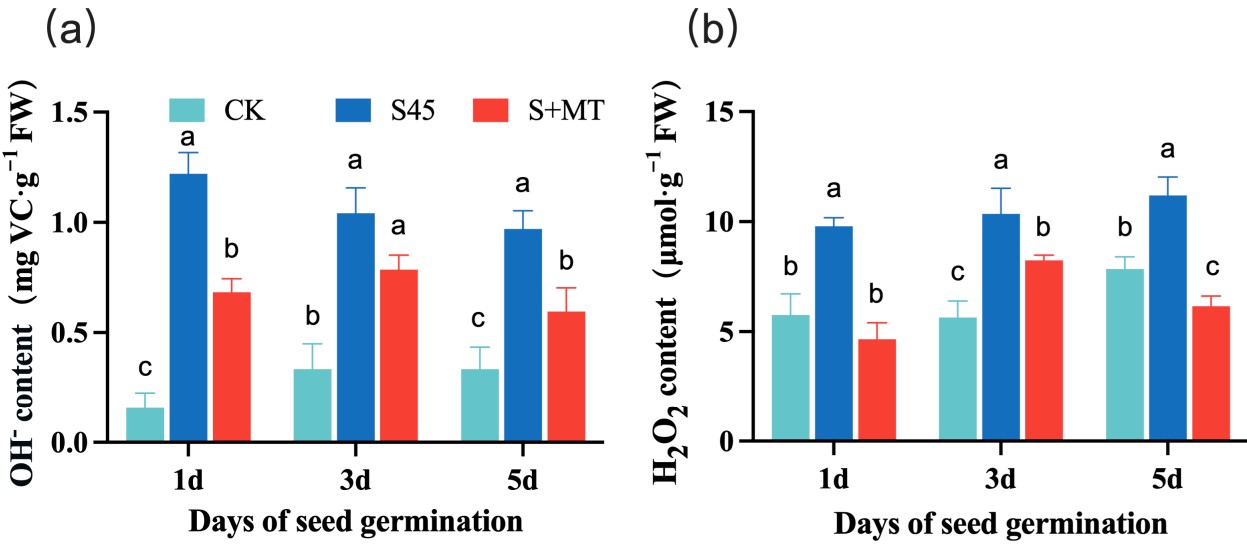

**Figure 4.** Effect of MT seed soaking treatment on OH$^-$ (**a**) and H$_2$O$_2$ (**b**) contents of oats under saline–alkali stress. CK is the control, S45 is saline–alkali stress, and S + MT is MT treatment + saline–alkali stress. Each lowercase letter represents the significance level of the difference between treatments ($p < 0.05$).

Excess H$_2$O$_2$ in the plant body causes oxidative damage to cells and produces toxic effects. Saline–alkali stress significantly increased the H$_2$O$_2$ content of oats compared with that of CK (Figure 4b), and the reduction in H$_2$O$_2$ content after MT treatment was significant. After 1 d of saline–alkali stress, the H$_2$O$_2$ content of the S + MT group was reduced by 18.99% and 110.12% compared with that of the CK and S45 groups, respectively ($p < 0.05$). After 3 d of saline–alkali stress, the H$_2$O$_2$ content of the S + MT group increased compared with the previous time period, but it was still lower than that of the S45 group. After 5 d of saline–alkali stress, the H$_2$O$_2$ content in the CK and S45 groups showed a slow increase compared with the previous time period, and the H$_2$O$_2$ content in the S + MT group showed an increase and then a decrease compared with the previous time; the H$_2$O$_2$ content was significantly lower than that in the CK and S45 groups by 21.51% and 81.82%, respectively.

3.2.5. Evaluation of MT under Saline–Alkali Stress Using Hierarchical Cluster Analysis and Correlation Analysis

Hierarchical cluster analysis showed (Figure 5) that the antioxidant enzyme activities (SOD, POD, CAT, and APX), nonenzymatic antioxidant activities (GSH and AsA), and osmoregulatory substance contents (soluble sugars, proteins, and proline) of oat seeds were significantly reduced and MDA, H$_2$O$_2$, and OH$^-$ were higher than normal under saline–alkali stress. Compared with saline–alkali stress, MT treatment effectively increased antioxidant enzyme and nonenzymatic antioxidant activities and osmotic substance contents, and it significantly reduced MDA, H$_2$O$_2$, and OH$^-$ contents.

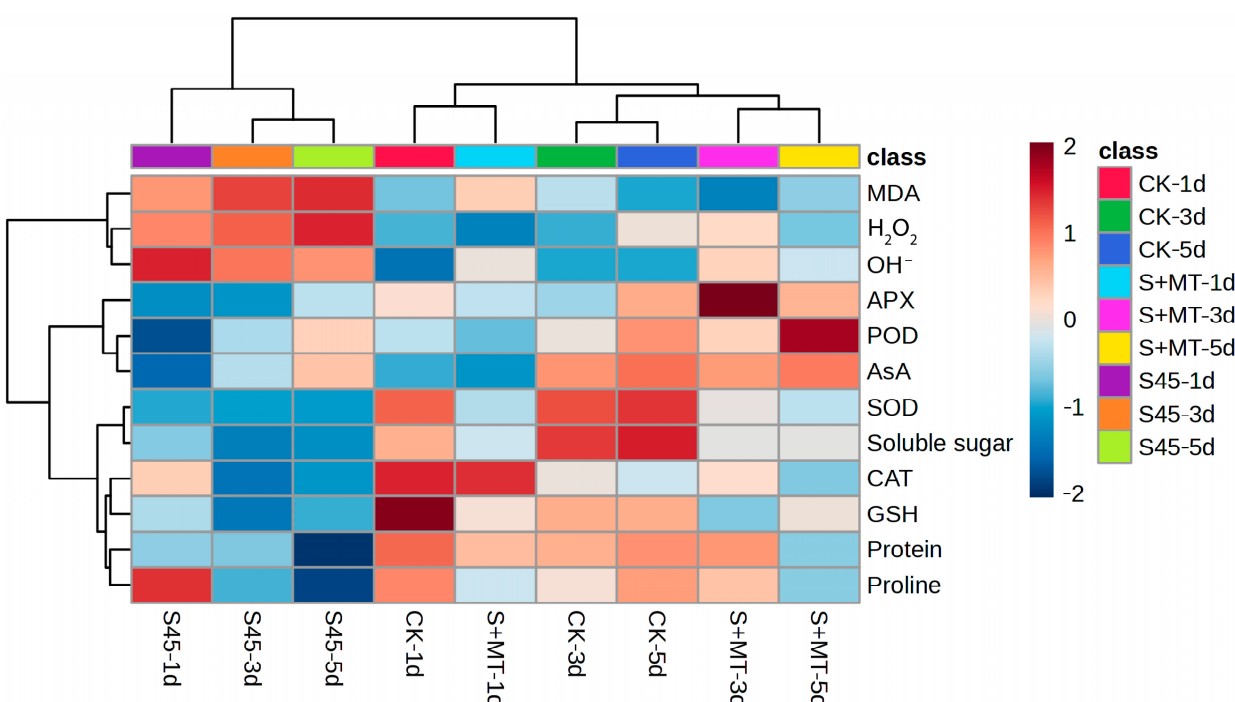

**Figure 5.** Hierarchical clustering analysis of different physiological indicators of oat seed germination under different treatments and different stress days after MT seed soaking treatment. CK is the control, S45 is saline–alkali stress, and S + MT is MT treatment + saline–alkali stress. The numbers 1, 3, and 5 after each treatment represent germination days 1, 3 and 5, respectively; the color intensity on the right shows their relative values.

Correlation analysis showed (Supplementary Figure S1) that 1 d after saline–alkali stress, there were highly significant and positive correlations ($p \leq 0.01$) for SOD with POD, GSH, AsA, and soluble sugars; there were highly significant negative correlations between SOD and MDA and OH$^-$. This indicates that the scavenging of ROS species during this period is mainly related to SOD, POD, GSH, and AsA activities.

After 3 d of saline–alkali stress (Supplementary Figure S2), the following highly significant and positive correlations were observed: SOD with GSH and soluble sugars, CAT with AsA, GSH with soluble sugars, AsA with soluble sugars, protein with proline, and H$_2$O$_2$ with OH$^-$ ($p \leq 0.01$). SOD, CAT, GSH, and AsA played the greatest roles in this period.

At 5 d after saline–alkali stress (Figure 6), the following highly significant positive correlations ($p \leq 0.01$) were observed: SOD with GSH, soluble sugars, protein, and proline; CAT with AsA; GSH with soluble sugars; MDA with H$_2$O$_2$ and OH$^-$; soluble sugars with protein and proline; and protein with proline. In addition, the following significant positive correlations were observed: CAT with soluble sugars, protein, and proline; APX with proline; GSH with protein and proline; AsA with protein; and H$_2$O$_2$ with OH$^-$ ($p \leq 0.05$). There were highly significant negative correlations between SOD and OH$^-$; GSH and H$_2$O$_2$ and OH$^-$; MDA and soluble sugars, protein, and proline; soluble sugars and OH$^-$; protein and OH$^-$; and proline and OH$^-$ ($p \leq 0.01$). The correlations between SOD and MDA; POD and H$_2$O$_2$; CAT and MDA and OH$^-$; APX and H$_2$O$_2$; and AsA and MDA, H$_2$O$_2$, and OH$^-$ were all significantly negative ($p \leq 0.05$), and the correlations between the other indicators were not significant ($p > 0.05$). The results above show that SOD and GSH played the most significant role at 5 d after stress.

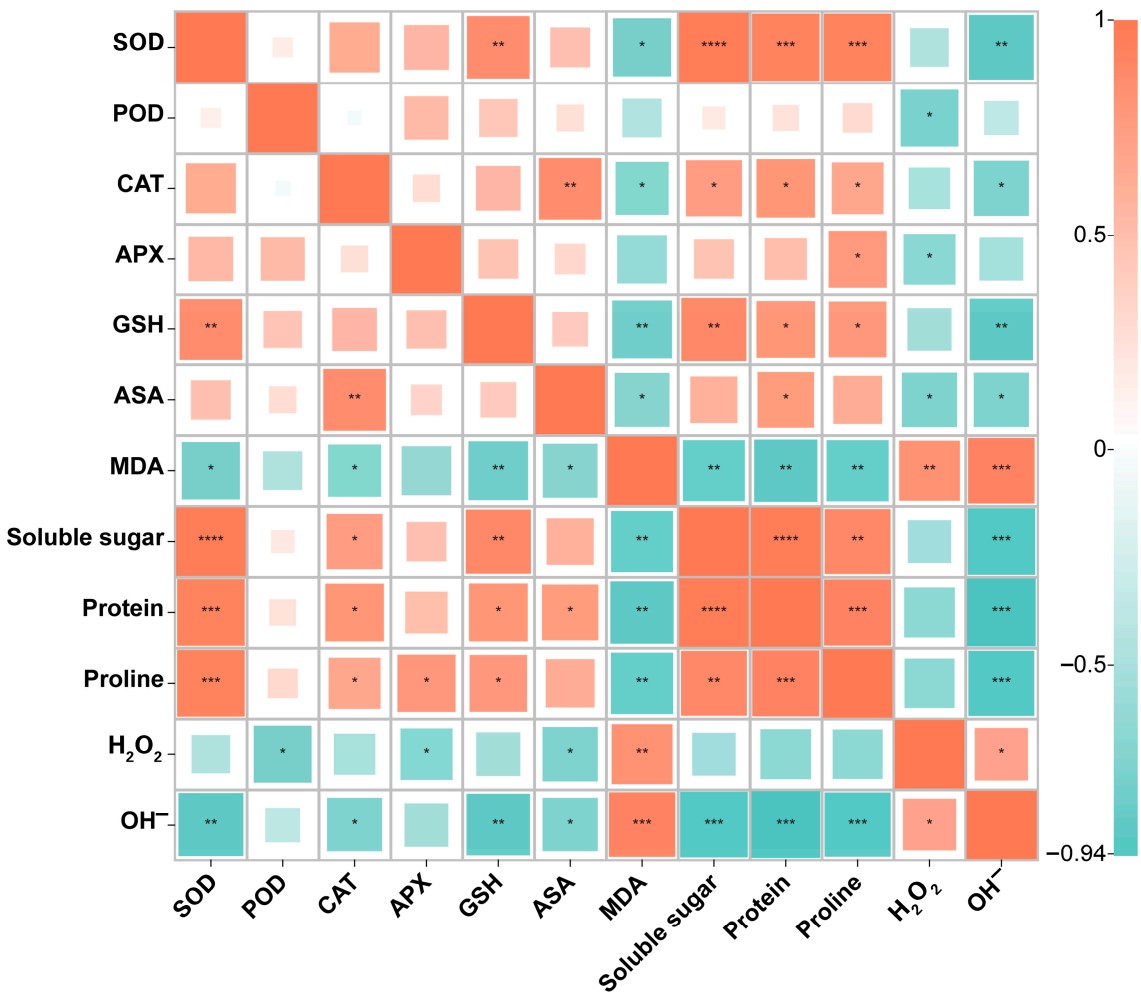

**Figure 6.** Correlation analysis of MT seed soaking treatments and physiological indicators of oats 5 d after saline–alkali stress. ****, ***, ** highly significant ($p \leq 0.01$), * significant ($p \leq 0.05$).

In summary, the degree of promotion of oat seed germination under saline–alkali stress by exogenous MT can be evaluated by SOD, POD, CAT, GSH, and AsA activities.

## 4. Discussion

The seed germination stage is the most intense activity period of plant life and the period when its resistance to stress is the weakest [41]. Numerous studies have shown that saline–alkali stress significantly inhibits seed germination in various plants [42–44]. In the early stage of seed germination, the strength of seed resistance is mainly assessed by germination indices such as germination rate, germination potential, and vigor index, whereas in the late stage of germination, resistance is evaluated primarily by morphological indices such as embryo length and root length [45]. Our experiments showed that a significant decrease in GR, VI, GL, RL, and FW compared with those of CK (Tables 1 and 2) under 45 mmol·L$^{-1}$ saline–alkali stress, which shows saline–alkali stress significantly inhibited oat seed germination. In numerous studies on plant stress resistance, it has been found that MT and its metabolic synthesis products play a pivotal role in plant resistance to adversity stress [46–50]. In this experiment, under the same saline–alkali stress, compared with the no MT treatment, the germination and morphological indices of oats showed a trend of increasing and then decreasing with increasing MT concentration, and the promotional effect of 24 h application was significantly higher than that of 12 h application, with the strongest effect under 100 μmol·L$^{-1}$ MT treatment (Table 3). The study shows that exogenous MT has a certain promotional effect on oat seed germination under saline–alkali stress, but this promotional

effect is affected by the application concentration and application time, demonstrating the phenomenon of "low promotion and high inhibition". This is consistent with the results of Xiao et al. [51] in their study on cotton. This may be due to the strong reducing ability of MT itself in plants, which can scavenge excessive ROS radicals in the body and maintain the metabolic balance of cellular ROS. In contrast, plants absorbed more MT from the external environment after exogenous application of high MT concentrations, and the excess MT in vivo induced oxidative denaturation of proteins in plants [52], which instead led to the accumulation of ROS species in vivo and inhibited the germination and growth of oat seeds. Previous study reported that 10–100 $\mu mol \cdot L^{-1}$ MT pretreatment mitigated the adverse effects of salt stress on seeds by reducing $H_2O_2$, MDA, and $Na^+$, and enhancing antioxidant enzyme activity and osmoregulatory substance content [53], whilst exogenous MT pretreatment increases seed germination rate, germination potential, and germination index under salt stress [54]. Our experiments showed that 100 $\mu mol \cdot L^{-1}$ MT seed soaking treatment for 24 h significantly increased GR, GP, GI, VI, RL, GL, FW, and DW, enhanced seed vigor, and promoted seed germination and growth in oat seeds under saline–alkali stress. This indicates that exogenous MT can alleviate the inhibition of oat seed germination by saline–alkali stress.

Saline–alkali stress alters the physiological, biochemical, and molecular processes in plants. Faced with the large number of ROS radicals generated under saline–alkali stress, the plant body adapts externally to this stressful environment by changing its morphological structure on the surface and internally by activating enzymatic and nonenzymatic antioxidant-regulated defense mechanisms to cope with stress at the cellular and molecular levels [55–57]. In the face of excessive ROS species accumulated in the body, SOD, which can breakdown the excessive ROS radicals in the plant into $H_2O_2$ and $O_2$, is the first line of defense in the plant antioxidant system. POD, CAT, and APX will then jointly break down and convert $H_2O_2$ into $H_2O$ and $O_2$ to scavenge ROS species. Antioxidants mainly include anthocyanins, GSH, AsA, and flavonoids, which are distributed in various parts of the cell to regulate the ROS species balance; GSH and AsA effectively scavenge ROS radicals by constituting an AsA-GSH cycle [58–60]. This study showed that saline–alkali stress significantly reduced the activities of SOD, POD, CAT, and APX in oat seeds, and the activities of POD, CAT, and APX increased at 5 d compared with 3 d as the stress time increased, whereas the SOD activity was always maintained at a lower level; the same phenomenon was found for GSH and AsA. This pattern may be related to increased membrane peroxidation due to elevated MDA, $H_2O_2$, and hydroxyl radical levels and excess ROS species accumulation, which inactivate antioxidant enzymes or inhibit their biosynthesis. Previous studies have shown that in addition to its ability to scavenge ROS species, MT can further scavenge excess ROS radicals in the body by regulating the activity of antioxidant enzymes in plants. This study showed that the activities of SOD, POD, CAT, and APX were significantly increased after the 100 $\mu mol \cdot L^{-1}$ MT seed soaking treatment of oat seeds; the antioxidant enzyme activities of seeds treated with MT were higher than those of seeds in the saline–alkali stress treatment in different periods of oat seed germination. The activity of POD increased gradually with the extension of the stress time, whereas SOD and APX increased and then decreased; at 5 d, except for CAT, the activities of SOD, POD, and APX were higher than those at 1 d. The increased activity of these antioxidant enzymes may be related to the fact that MT promotes the expression of antioxidant enzymes and nonenzymatic antioxidant-related synthetic genes in oat seeds, thus increasing the biosynthesis of antioxidant enzymes [50]. APX, the primary enzyme class of the AsA-GSH cycle, is the catalyst in the AsA-GSH conversion cycle. In this study, we found that the GSH and AsA contents of oats peaked at 5 d after MT treatment and were higher in either period of germination compared with those of the S45-treated group. Correlation analysis also showed (Figure 6) that GSH and AsA were significantly and positively correlated with the decreases in MDA and $OH^-$ contents ($p \leq 0.05$). The higher APX, GSH, and AsA contents ensure the AsA-GSH cycle in oats, which prevents the formation of membrane peroxisomes, maintains stability and stabilizes protein [61]. This result was also obtained in

previous studies, and exogenous application of MT significantly improved the ROS species content and increased the activities of intracellular antioxidant enzymes, thus enhancing the tolerance of soybean to saline–alkali stress [62]. Similar findings were also found in MT studies on cucumber [63] and tomato [64], where the exogenous application of MT significantly enhanced their antioxidant enzyme activities under saline stress and promoted seed germination and seedling growth. It is suggested that exogenous MT can mitigate the saline–alkali stress injury suffered by oat seeds by increasing the activity of antioxidant enzymes and nonenzymatic antioxidants and by affecting the AsA-GSH cycle.

Plants under saline–alkali stress have antioxidant defense mechanisms and rely on osmoregulatory mechanisms to ensure water uptake by plants. Studies have shown that osmotic stress typically occurs in the early stages of saline–alkali stress [65], that plants under saline–alkali stress can induce the accumulation of osmotic substances such as proline and soluble sugar content to resist saline stress, and that there is variability among different species [66,67]. This study showed that oat seeds under saline–alkali stress had lower contents of osmotic substances such as protein, soluble sugar, and proline, and all showed a gradual decrease with increasing stress duration (Figure 3). This indicates that the osmotic stress response of oats was disrupted in this environment and that the high pH environment disrupted the ionic homeostasis of cells, causing a decrease in cellular osmotic potential and membrane stability, thus affecting the normal physiological response of oats. Correlation analysis also indicated that protein, soluble sugar, and proline contents were closely correlated with increased MDA content ($p \leq 0.05$). In a previous study, MT-treated seedlings had higher chlorophyll, soluble sugar, proline, and soluble protein contents; a significantly increased net photosynthetic rate and transpiration rate; and higher carbon assimilation enzyme activities than normal seedlings, all of which indicate that exogenous application of MT can reduce stress damage induced by saline–alkali stress by increasing the photosynthetic capacity and osmotic substance content of cucumber seedlings [68]. Studies by Kamiab [69] and Ding [70] similarly illustrated that MT could effectively promote the accumulation of osmoregulatory substances, thereby improving the osmoregulatory capacity of cells and increasing the water content of plant tissues. Our study also supported this conclusion. Exogenous application of 100 μmol·L$^{-1}$ MT significantly increased the contents of protein, soluble sugar, and proline in oat seeds, and the contents of protein, soluble sugar, and free proline showed a trend of increasing and then decreasing with the duration of saline–alkali stress, reaching a peak at 3 d. This indicates that the osmoregulatory mechanism of oat seeds was active, and promoted the accumulation of osmoregulatory substances in oats. Similar findings have been verified in wheat [71] and rice [72], and the results may be related to MT increasing the synthesis of intracellular organic solvents, which prevented water loss from the cellular protoplasm, alleviated the damage caused by a saline–alkali stress environment to oat seeds, and promoted seed germination.

When saline–alkali stress reaches a certain level and time, the antioxidant mechanism in plants is broken, and many ROS species (e.g., $H_2O_2$ and $OH^-$) will accumulate, thus destroying the selective permeability of the cell membrane and the extravasation of many electrolytes, causing oxidative cellular damage and even programmed death [4,73]. We found that with the extension of saline–alkali stress time, the MDA and $H_2O_2$ contents of oats showed a gradual increase, and the $OH^-$ content was slightly decreased, but the contents of all three were higher than those of the CK and S + MT treatment groups in either period (Figures 3 and 4), which induced membrane peroxidation. It has been shown that MT, as an amphipathic molecule, and its two side chain groups (the 5-methoxy group and 3-amide group) play a crucial role in scavenging ROS [74]. Among ROS, $OH^-$ is the most vigorous class known, and the mechanism of MT-mediated $OH^-$ scavenging mainly involves the addition of $OH^-$ to the C3 of the indole ring, causing cyclization between the C2 position of the indole ring and the N on the ringside chain, which triggers its reaction with a second OH- to form cyclic 3-hydroxy MT and $H_2O$ [75]. In this experiment, oat seeds treated with 100 μmol·L$^{-1}$ MT significantly reduced the contents of MDA, $H_2O_2$,

and $OH^-$ compared with water-soaked seeds, and these contents were lower than those of the S45-treated group in either stress period. It is evident that exogenous MT induces the removal of excess $H_2O_2$ and $OH^-$, and the lower $OH^-$ content inhibits MDA production, thus protecting the selective permeability and membrane stability of cell membranes and alleviating cellular oxidative damage caused by saline–alkali stress. Correlation analysis similarly indicated that MDA was significantly and positively correlated with $H_2O_2$ and $OH^-$ contents ($p \leq 0.05$). This result is consistent with previous reports that abiotic stress significantly increased MDA, $H_2O_2$, and superoxide anion ($O^{2-}$) in rice, whereas MT application suppressed MDA and $H_2O_2$ contents as well as $O^{2-}$ productivity while enhancing antioxidant enzyme activity and improving drought tolerance in rice [72]. Studies on cotton also found that salt stress promoted the accumulation of $H_2O_2$, MDA, organic osmotic substances, and inorganic osmotic substances. MT application simultaneously increased endogenous MT, soluble sugar, protein, proline, and $K^+/Na^+$ contents, suggesting that MT can effectively ameliorate the inhibition of cotton seed germination by salt stress by enhancing osmotic substances and regulating ionic homeostasis [53]. The present study also further confirms the important role played by MT in scavenging ROS.

## 5. Conclusions

Under saline–alkali stress, different concentrations of MT treatment showed "low promotion and high inhibition" of the germination of oat seeds, with the strongest effect under the 100 $\mu mol \cdot L^{-1}$ MT treatment. Analysis of seed physiological activity showed that MT could promote seed germination by enhancing antioxidant enzyme activity, accelerating the AsA-GSH cycle, increasing the contents of osmoregulatory substances, and scavenging excess $OH^-$ and $H_2O_2$ produced by saline stress, while reducing MDA content, decreasing cellular oxidative damage, and protecting selective permeability of the cell membrane.

**Supplementary Materials:** The following supporting information can be downloaded at: https://www.mdpi.com/article/10.3390/agronomy13051327/s1, Figure S1: Correlation analysis of MT seed soaking treatments on physiological indicators of oats 1 d after saline-alkali stress; Figure S2: Correlation analysis of MT seed soaking treatments on physiological indicators of oats at 3 d after saline-alkali stress.

**Author Contributions:** Q.W. and W.X. performed the experimental design, investigation, formal analysis, and writing. C.R. performed supervision, review, validation, and project management. C.Z. and C.W. performed data validation, analysis, and visualization. J.L. and Q.R. performed material collection, investigation, and review. X.L., L.W. and D.X. contributed to the review and editing. L.G. and J.W. performed the review, supervision, editing, conceptualization, investigation, project management, and financial support. All authors have read and agreed to the published version of the manuscript.

**Funding:** China Agriculture Research System funded this study, Grant No. (CARS-07-B-2).

**Data Availability Statement:** The all datasets supporting the conclusions of this article are included within the article. If not included in the manuscript are available from the corresponding author upon reasonable request.

**Acknowledgments:** This work was supported by the China Agriculture Research System (CARS-07).

**Conflicts of Interest:** The authors declare no conflict of interest. All authors have read and agreed to the published version of the manuscript.

## Appendix A

**Table A1.** Nutrient composition of the test-modified Hoagland nutrient solution.

| Composition | mg·L$^{-1}$ |
|---|---|
| Ca (NO$_3$)$_2$·4H$_2$O | 945 |
| KNO$_3$ | 506 |
| NH$_4$NO$_3$ | 80 |

**Table A1.** *Cont.*

| Composition | mg·L$^{-1}$ |
|---|---|
| KH$_2$PO$_4$ | 126 |
| MgSO$_4$ | 241 |
| FeNaEDTA | 36.7 |
| KI | 0.83 |
| H$_3$BO$_3$ | 6.2 |
| MnSO$_4$·H$_2$O | 16.9 |
| ZnSO$_4$·7H$_2$O | 8.6 |
| Na$_2$MoO$_4$·2H$_2$O | 0.25 |
| CuSO$_4$·5H$_2$O | 0.025 |
| CoCl$_2$·6H$_2$O | 0.025 |
| Totals | 1977.53 |
| pH (25 °C) | 5.8 ± 0.2 |

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
