# Peer review of "Physiological and Biochemical Mechanisms of Exogenous Melatonin Regulation of Saline–Alkali Tolerance in Oats"

_agronomy, doi:10.3390/agronomy13051327_

Round 1

Reviewer 1 Report

The manuscript agronomy-2373614 entitled “Physiological and biochemical mechanisms of exogenous melatonin regulation of saline-alkali tolerance in oats” aimed to evaluate the effect of seed soaking with melatonin in different application time and concentration on seed germination index and plant defense response under salt-alkali stress.

The author reported that different concentration and application time improved the seed germination, and the 100 100 μmol·L-1 melatonin showed the best results.

The manuscript is well written and designed that could be published after minor revision.

The abstract has too long sentence (Line: 22-26; 26-32) and I suggest to reformulated.

Abbreviation must be defined at first mentioned, please revise the introduction.

Preferably to prove the sensitivity of the used variety to salt-alkali soil with previous experiments or considerable reference.

I suggest to re-present the first experiment 16 treatments in Table, to be more clear.

The seed germination index calculation procedure must be supported by considerable references.

Line 282-285: I suggest to remove this sentence as it already showed in Table.

Line 364 also 379; this sentence could be moved to discussion part.

Some used references are very old, which should be updated, if applicable.

Nothing

Reviewer 2 Report

agronomy-2373614-peer-review-v1

 Article: Physiological and Biochemical Mechanisms of Exogenous Melatonin Regulation of Saline-alkali Tolerance in Oats.

The idea of the manuscript is unique, new, and complete for publication, with minor corrections

Abstracts: good.

Keywords: add antioxidant enzyme activity.

Introduction:

The introduction was written in a unique way.

Materials and Methods:

What is the authors' evidence for the sensitivity of an oat cultivar to salinity?

Is replicates only one Petri dishes of 30 seeds ? (Experiment 2: five Petri dishes per replicate).

2.3.2. Morphological and Biomass Indicators, the authors reported the use of 10 seedlings, are they seedlings, and if so, at what age? , wherever you plant it in Petri dishes ?

A Comprehensive Evaluation of Different MT Treatments to Improve the Saline alkali Tolerance of Oats, at any age too.

In the second experiment on what basis the authors used concentration a 100 μmol・L-1 MT solution ? Is it just the best treatment?

The experiment was conducted from July 2022 to November 2022 in the greenhouse, a large period of time on two experiments, the first experiment seven days and the second it five days, as you mentioned, please clarify.

Results

Line 270-285:  the paragraph needs to be rewritten, focusing on the different transactions, and there is no need to arrange all the transactions.

Discussion: Presented appropriately with the manuscript, with the possibility of abbreviation.

Conclusions: good section.

Reviewer 3 Report

The reviewed paper is well-written and interesting in terms of the mechanism of melatonin action under stress conditions. 

I have four major comments:

1) All tables and figures should be understandable to the reader without reference to the text (please explain all abbreviations)

2) In the present form Tables 1, 2, and 3 are difficult to follow, especially in terms of statistical analysis; please reformat them or break them into smaller ones

3) out of figures 5 to 7, I suggest leaving only the most important one in the main text, and moving the rest to the supplement 

4) I suggest shortening the discussion section by omitting very obvious facts

Additional minor comments:

- Please change the '1st d.' to '1st day'

- there are missing data for the unit used in Figures - is it g FW or DW?

- use ROS for reactive oxygen species in the text (especially in the Discussion section)
